# The Role of Needle Fear in Pediatric Flu Vaccine Hesitancy: A Cross-Sectional Study in Bologna Metropolitan Area

**DOI:** 10.3390/vaccines10091388

**Published:** 2022-08-25

**Authors:** Esther Rita De Gioia, Adalisa Porqueddu, Ornela Nebiaj, Alessandro Bianconi, Alice Conni, Marco Montalti, Paolo Pandolfi, Renato Todeschini, Maria Pia Fantini, Davide Gori

**Affiliations:** 1School of Hygiene and Preventive Medicine, Department of Biomedical and Neuromotor Sciences, Public Health and Medical Statistics, University of Bologna, Via San Giacomo 12, 40126 Bologna, Italy; 2Department of Public Health, Bologna Local Health Authority, Via Gramsci, 12, 40124 Bologna, Italy; 3Unit of Hygiene, Department of Biomedical and Neuromotor Sciences, Public Health and Medical Statistics, University of Bologna, Via San Giacomo 12, 40126 Bologna, Italy

**Keywords:** vaccine hesitancy, needle fear, flu vaccine, children, survey, SAGE group

## Abstract

(1) Background: vaccination is the most effective way to prevent influenza and reduce its complications. The main aim of the study is to assess a possible increase of parents’/caregivers’ pediatric flu vaccination adherence due to a nasal administration as an alternative to injection in Bologna. (2) Methods: 169 parents/guardians of children who were joining the COVID-19 pediatric vaccination session in Bologna were interviewed. The results were summarized using descriptive statistics. A multiple logistic regression model was used to assess the determinants of the change in flu vaccine uptake if offered without injection administration. All analyses were conducted using STATA and R-Studio software. (3) Results: Only 29.0% of parents were informed about pediatric flu vaccination by pediatricians, and 32.5% heard about pediatric flu vaccination. Almost 72.2% of parents declared that they would not have their children vaccinated against influenza. Thus, 40.2% of them changed their opinion after being informed about the existence of a non-injective vaccine. Needle fear in children turned out to be a determinant of this opinion change (OR = 3.79; 95% CI = 1.63–9.43; *p* = 0.003). (4) Conclusions: the study has confirmed that needle fear is a determinant of vaccine hesitancy and that a different method of administration may increase parents’/guardians’ adherence.

## 1. Introduction

Flu is a contagious viral infectious disease caused by influenza viruses, which belong to the single-stranded RNA genome family Orthomyxoviridae [1]. Influenza virus attacks the respiratory system; it can cause mild to severe symptoms and life-threatening complications, including death, even in children in good health and adults [2]. Influenza viruses spread through infected patients’ saliva droplets released from cough or through normal breathing, so the role of asymptomatic patients is fundamental [3]. Influenza is usually self-limiting in individuals in good health [4]. However, children, in particular those younger than 2 years old, are under a higher risk of developing serious flu-related complications, such as primary viral pneumonia or secondary bacterial pneumonia [3,5]. The best way to prevent flu is vaccination [5], which offers the best defense against its potential consequences and can reduce its spread. The benefits of having the vaccine are as follows: reduction of flu illnesses, of doctor’s visits and of sick days either at work or school; moreover, it reduces the risk of flu-related hospitalization and death in children [6].

The Centers for Disease Control and Prevention recommend that every child get a flu vaccine every year, ideally by the end of October, starting when he/she is 6 months old [6]. In Italy, flu vaccination is strongly recommended for people over sixty years old, pregnant women and patients in postpartum status, long-term hospitalized patients, patients with chronic diseases and specific groups of workers [7]. Since 2019, the Italian Ministry of Health recommended the flu vaccination even to children between six months and six years old, in order to reduce influenza virus propagation in the context of the SARS-CoV-2 pandemic emergency. Babies and children are the most affected by influenza, and they represent one of the main influenza drivers for the entire population. Between 2010 and 2020, 60.7% of 6/10-year-old Italian children got infected by influenza virus [3]. The Italian vaccination coverage rate for influenza in the 5/8-year-old population was 3.1% after the 2019–2020 campaign. One year later, after the 2020–2021 campaign, the rate of vaccination coverage against influenza among 5/8-year-old was 13.1%, highlighting a positive trend in flu vaccination coverage in the pediatric population in Italy [8].

Nowadays, in addition to Injectable Influenza Vaccine (IIV4), the Live Attenuated Influenza Vaccine (LAIV4) is also available. LAIV4 is given as a nasal spray that is squirted up each nostril, and it is currently approved for use in 2/49-year-old non-pregnant people in good health [9]. Tetravalent vaccine FLUENZ TETRA by AstraZeneca AB was approved by the European Medicines Agency with the European Commission deliberation (2013)/8894 cor. in 4 December 2013 [10].

Despite evidence of vaccine safety and efficacy, vaccine hesitancy (VH) is a pervasive and context-specific phenomenon that is spreading more and more. In 1999, the World Health Organization (WHO) established the Strategic Advisory Group of Experts (SAGE), which provides guidance on vaccine and immunization activities [11]. In 2014, the SAGE Working Group on Vaccine Hesitancy (established in 2012) developed a model to categorize the factors that affect the decision to accept, delay or reject the vaccine. The Working Group defined “*the delay in acceptance or refusal of vaccination despite availability of vaccination services*” as vaccine hesitancy. Therefore, hesitancy might be considered as a spectrum extending from those who totally accept the vaccination to those who completely refuse it, and there is a full range of middle positions in between [12].

In 2001, SAGE displayed a discrepancy between scientific evidence and perception of risk in hesitant individuals and proposed a strategy to confront the population’s concern about AEFI (Adverse Event Following Immunization) [11]. VH is influenced by several elements, such as concern about vaccine safety, fear of needles, lack of awareness about vaccine-related preventable disease and trust in health-care workers [12]. All these elements can be summarized by the 5Cs model: the Cs stand for Complacency (level of risk perception about potentially avoidable diseases and the importance of getting immunized); Confidence (trust in vaccine safety and efficacy and in the authority of the health system); Convenience (perception of the quality of vaccination services); Collective responsibility (perception of the importance of protecting others); and Calculation (obtaining exhaustive information before deciding whether to have a vaccination) [13].

Furthermore, another element of hesitancy is represented by the fear of needles [14]. The term “needle fear” describes the feeling of anxiety related to situations where needles or injections are used [14], while “needle phobia” is described by the Diagnostic and Statistical Manual of Mental Disorders (DSM-V) as being part of specific phobias of a blood-injection-injury type [15]. Needle phobia often occurs with generalized fear; it frequently develops as a diphasic vasovagal response with a decrease in blood pressure, which can end in fainting [14]. The reported prevalence rates of needle phobia strongly diverge in the literature from 3.5% to 20%, and findings are often conflicting. Estimating the number of patients who may avoid vaccination due to needle phobia is difficult because this group is known to avoid contact with the medical system [15]. The fear of needles shows a considerable variation in prevalence by age: the majority of children exhibited needle fear, while estimates in adolescents are 20–50% and 20–30% in young adults. In general, needle fear decreases with increasing age. The fear of needles often causes avoidance of preventive medical measures. In particular, the fear of needles expresses itself in the high prevalence of vaccination avoidance, especially for influenza, but similar results were collected for the avoidance of other vaccines (such as pneumococcal and tetanus vaccination) [14].

Given these premises, increasing adherence to flu vaccination in the pediatric population appears to be a relevant public health concern. The evaluation and implementation of solutions that may decrease flu VH are crucial in order to face this concern. Thus, the aim of our study (named “Vacci Senz’Ago!”) was to evaluate if the proposal of a non-injective route of administration may represent a possible encouragement to have children vaccinated.

## 2. Materials and Methods

This cross-sectional study investigated parents’/guardians’ hesitancy rate related to influenza vaccination. During April 2022, parents/guardians were enrolled at the COVID-19 Vaccination Hub managed by the Bologna Local Health Authority (LHA). All participants were parents or guardians who were accompanying their children to get the COVID-19 vaccine. Other eligibility criteria were parents’/guardians’ residence/domicile in the Bologna Metropolitan Area and their children being aged >2 and <14 years old.

Participants were surveyed through an anonymous questionnaire. The questionnaire was generated using Google Forms and was filled autonomously or with the help of a medical doctor member of the research team. No randomization or special selection was carried out. All participants provided informed consent to being anonymously included in the database for study participation.

The study protocol was reviewed and approved by the Ethics Committee for Research and Experimentation of the University of Bologna with the approval number 0087724 of the 26 April 2022.

The variables collected include socio-demographic variables (e.g., gender, age, educational stage, etc.), background vaccination rates among children, parents’ perception of injection invasiveness, parents’ awareness about pediatric flu vaccination and parents’ perceptions of a non-injective vaccine (e.g., nasal administration, etc.). The English version of the questionnaire can be found in the Appendix A.

All data were collected anonymously, with a unique identification number being assigned to each participant.

Qualitative variables were described as absolute and relative frequencies; quantitative variables were described as means with standard deviations.

Determinants of the change in flu vaccine uptake if offered without injection administration were assessed by multiple logistic regression analysis. Participants who responded that they would get their children vaccinated in the future were excluded from this analysis. A backwards elimination process was used to choose the variables to be included in the final regression model (The final regression model output can be found in Appendix A).

The results of the multiple logistic regression analyses were presented as odds ratios (ORs) with 95% Confidence Intervals (95% CI). Data was collected using Microsoft Excel (Microsoft Corporation, Redmond, WA, USA). All analyses were carried out using Stata Statistical Software 15 (StataCorp, College Station, TX, USA) and R-Studio (RStudio, PBC, Boston, MA, USA).

## 3. Results

### 3.1. Population Characteristics

In the course of the data collection, 169 participants were recruited among parents/guardians who accompanied their children to receive the COVID-19 vaccine. A total of 59.2% of them were female (*n* = 100). The mean parents’ age was 42.0 (S.D. = 6.8). The mean children’s age was 8.1 (S.D. = 2.2). The educational stage among parents was heterogeneous, with 38.5% (*n* = 65) of them having a middle school degree or less, 27.2% (*n* = 46) having a high school degree and 34.3% (*n* = 58) having a university degree (see Table 1).

### 3.2. Questionnaire Descriptives

Among surveyed parents, 107 (63%) answered that their children have relatives older than 60 years old; 8 children (7.5%) never spend time with those relatives, 46 (43.0%) spend time with those relatives once a week or less, 22 (20.6%) spend time with those relatives more than once a week, but less than one hour each day, and 31 (29.0%) spend time with those relatives at least one hour every day.

Furthermore, 54 (32.0%) parents responded that their children have relatives with at least one chronic disease; 5 (9.3%) never spend time with those relatives, 15 (27.8%) spend time with those relatives once a week or less, 12 (22.2%) spend time with those relatives more than once a week, but less than one hour each day, and 22 (40.7%) spend time with those relatives at least one hour every day. Additionally, 7 (4.1%) children had chronic disease (see Table 2).

Only 9 (5.3%) children were vaccinated against influenza during the 2021/22 campaign. Moreover, 165 (97.6%) parents declared that their children got all mandatory vaccinations, and 102 (60.4%) of them said that their children manifested concern or fear of injection in the past.

Among surveyed parents, 49 (29.0%) of them answered that their children’s pediatricians had advised them to have their sons/daughters vaccinated against influenza; 55 (32.5%) parents were not aware that flu vaccination is recommended among the pediatric population. Acknowledging this, only 47 (27.8%) declared that they will have their children vaccinated against influenza (see Table 3).

However, 49 (40.2%) parents who responded that they would not have their children vaccinated against influenza in the future, declared that they would consider the vaccination if administered without a needle injection. Furthermore, 39 (83.0%) parents who responded that they would have their children vaccinated against influenza in the future, declared that they would prefer a no-needle vaccination (See Table 4).

### 3.3. Regression Model Results

Among parents who would not get their children vaccinated against influenza in the future, those who had noticed their children fearing injection during past vaccinations turned out to be more likely to consider joining a pediatric flu vaccination campaign in the future if it would involve a non-injective administration (OR = 3.79; 95% CI = 1.63–9.43; *p* = 0.003).

## 4. Discussion

In the sample, needle fear has been found to be a determinant of adherence to future flu non-injective vaccination pediatric campaigns in hesitant parents/guardians. Injection fear is a determinant of VH in the pediatric population, as declared by the WHO SAGE group [12]. This phenomenon particularly affects children, and it decreases with age [14]. In this view, it is essential to suggest solutions that involve this specific population strata. Indeed, the proposal of a non-injective route of administration led to a potential change of opinion in parents/guardians who declared that they would not have their children vaccinated against influenza in the future campaign. The study results show how a flu vaccination administered without needles, e.g., nasal spray, might be more accepted by parents/guardians for their children and might thus increase the level of adherence.

Additionally, the study shows that more than one-third of parents never heard about the existence of the pediatric flu vaccine. The low flu vaccine awareness rate is congruent with several studies. For instance, a 2016 Australian cross-sectional study of 539 people found that only 33% of parents were aware of children’s influenza vaccine recommendation [16]. Therefore, a structured communication campaign should be integrated into any immunization campaign, and there are several evidences [17,18,19,20] that lack of communication may create serious obstacles and compromise an immunization intervention. Information should be developed by healthcare providers, and communication experts, mass electronic media, digital media and social mobilization may represent effective tools; however, they should be strictly selected and monitored so as to refine their impact [21,22].

Moreover, less than one-third of parents/guardians who were surveyed declared that their children’s pediatricians recommended a flu vaccination. As shown in the literature, physicians’ recommendation is a key factor in the parental decision-making process to vaccinate their children [23]. On the one hand, one of the most common reasons why parents do not have their children vaccinated against influenza is that they did not consider it, showing a lack of risk awareness for the disease and/or disinformation about the possibility of countering VH. On the other hand, one of the most important factors for parents in making the decision to vaccinate their children is the recommendation from their healthcare provider [24]. Parents frequently cite their pediatrician as a trusted source of information regarding vaccines [25].

Pediatric healthcare providers should make influenza vaccines easily accessible for all children. Sending alerts to parents/guardians could be a way to achieve this result. Administering the influenza vaccine during child examinations and sick visits as well as in hospitalized patients, especially those at a high risk of influenza complications, may represent another option to increase the adherence [26].

Finally, considering parents’/guardians’ low awareness of the vaccine, there is the need to improve and strengthen the promotion of flu vaccine campaigns. Public Health Institutions should work on a way to deliver vaccination information, for example by providing univocal messages and advice: this should be done in synergy with pediatricians, who remain the main point of reference for parents/guardians about children’s health.

### Study Limitations

Due to the fact that all study participants were enrolled in a COVID-19 vaccination hub, a selection bias during the collection of the samples may not be excluded. As a matter of fact, since parents/guardians were voluntarily leading their children to be vaccinated against COVID-19, the influenza VH might have been underestimated.

Another limitation of this study is that the sample size is small. For this reason, the power of logistic regression statistics is limited, and the external generalizability of the results might thus be affected.

## 5. Conclusions

The “Vacci Senz’Ago!” study highlighted the fact that needle fear is a determinant of a hypothetical adherence to future flu non-injective vaccination pediatric campaigns in hesitant parents/guardians in the Metropolitan Area of Bologna.

Contrasting VH is an important theme for local public health policymakers, who should try to find out the best way to increase vaccine adherence by studying the most widespread barriers in the population and try to reduce them as much as possible.

The proposal of a vaccination through an alternative, non-injective route of administration, such as through a nasal spray, seems to be a factor in reducing hesitancy in parents who are not considering having their children vaccinated in the future. Furthermore, it might represent an incentive for parents who are already inclined to have their children vaccinated.

## Figures and Tables

**Table 1 vaccines-10-01388-t001:** Socio-demographic variables.

Characteristics		N (%) OR Mean (±S.D.)
Parents’ gender	M	69 (40.8) *
	F	100 (59.2) *
Parents’ age		42.0 (6.8) **
Children’s age		8.1 (2.2) **
Parents’ educational stage	Middle school degree or lower	65 (38.5) *
	High school degree	46 (27.2) *
	University degree	58 (34.3) *

* N (%), ** Mean (±S.D.).

**Table 2 vaccines-10-01388-t002:** Fragilities in children and relatives.

Characteristics		N (%)
Relatives over 60	Yes	107 (63.3)
	No	62 (36.7)
Time spent with relatives over 60	Never	8 (7.5)
	Once a week or less	46 (43.0)
	More than once a week	22 (20.6)
	At least one hour a day	31 (29.0)
Relatives with chronic disease	Yes	54 (32.0)
	No	115 (68.0)
Time spent with relatives with chronic disease	Never	5 (9.3)
	Once a week or less	15 (27.8)
	More than once a week	12 (22.2)
	At least one hour a day	22 (40.7)
Children with chronic disease	Yes	7 (4.1)
	No	162 (95.9)

**Table 3 vaccines-10-01388-t003:** Past vaccination experiences and future perspectives.

Characteristics		N (%)
Influenza vaccination during 2021/22 campaign	Yes	9 (5.3)
	No	160 (94.7)
All mandatory vaccinations	Yes	165 (97.6)
	No	4 (2.4)
Manifested fear of injection during prior vaccinations	Yes	102 (60.4)
	No	67 (39.6)
Pediatrician advised influenza vaccination	Yes	49 (29.0)
	No	120 (71.0)
Parents’ awareness about pediatric influenza vaccine	Yes	55 (32.5)
	No	114 (67.5)
Influenza vaccination in the future	Yes	47 (27.8)
	No	122 (72.2)
Would consider non-injective vaccination	Yes	88 (52.1%)
	No	81 (47.9%)

**Table 4 vaccines-10-01388-t004:** Cross-table between “Influenza vaccination in the future” and “Would consider non-injective vaccination” variables.

	Would Consider Non-Injective Vaccination	Would Not Consider Non-Injective Vaccination	
Would not consider influenza vaccination in the future	49 (40.2%)	73 (59.8%)	122 (100%)
Would consider influenza vaccination in the future	39 (83.0%)	8 (17.0%)	47 (100%)
	88 (52.1%)	81 (47.9%)	169 (100%)

Chi-squared without Yates correction; Chi squared equals 24.921 with 1 degree of freedom; The two-tailed *p* value is less than 0.0001.

## Data Availability

The dataset generated and analyzed during the current study can be made available by the corresponding author, A.C., on reasonable request. E.R.D.G., A.P., O.B., A.B., A.C., M.M. and D.G. are responsible for and accessed the raw data involved in the study.

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
