# Peer review of "The Role of Needle Fear in Pediatric Flu Vaccine Hesitancy: A Cross-Sectional Study in Bologna Metropolitan Area"

_vaccines, 2022, doi:10.3390/vaccines10091388_

Round 1

Reviewer 1 Report

Estimated Authors,

thank you for the opportunity to review this very interesting cross-sectional study on the determinants of potential VH towards SIV through nasal administration. According to this small, but well designed and described study on parents and guardians of children performing COVID-19 vaccination, the main effector of opinion change may be identified  in the needle fear in children (and not only in children, honestly...), with a multivariable OR of 3.79, 95%CI 1.63-9.43. This result, even though it should be acknowledged as minimally generalizable because of the small sample size (in fact, acknowledging a pre-testing prevalence of VH of 50% among sampled, nearly a double sample size of 384 participants could be requested), is particularly interesting as it stresses how important could be improving the delivery and the referral of spray-vaccination for SIV as it comes commercially available. A future iteration of this research could assess the acceptance of this vaccination in areas (e.g. Romagna subregion of Emilia Romagna, the very same area of this study) historically characterized by high rates of VH.

In my opinion, the present paper could be accepted as it is (only I would suggest to avoid the "sigma" notation for S.D., as most of the reader could ignore the equivalence between the two notations..., but it is only a personal consideration).

Author Response

Dear reviewer,

please see the file attached.

Reviewer 2 Report

The paper describes a well known reality, that of VH on which a great deal of literature exists (see among other, D'Errico et al, Vaccine, doi: 10.3390/vaccines9020110.). However, the specific issue of needle fear dealt in the paper, makes it quite interesting and may contribute to counteract VH. 

I suggest to modify the title, as "vacci Senz'ago" is understable only for Italian readers and it can make the paper of lesser immediately impact for international readers. 

Furthermore, the sentence lines 73-75 (pag. 2) "Therefore, hesitancy might be described as a spectrum that extends from those who accept the vaccination to those who refuse it, and there are a lot of positions in between [12]", is not clear in its meaning; please better clarify.  

Author Response

(The authors gave the same response as above.)

Reviewer 3 Report

This study is focused on the preference of administration modality of pediatric flu vaccination. The topic is of interest.

Title - Pls translate somewhere in the text the part in the title which is in Italian. Consider the title which should be communicative to an international audience.

Under introduction.

L 60 " Despite the evidence of complication..." – did the authors mean ?no complications"?

Start a new paragraph in l .68.

Make sure that words of causality (influence, factors) are used only in reporting on experimental studies, and otherwise use words of association. E.g., " is influenced" when the data is observational.

Please cite literature on administration modality of medication. There are a number of long term medical conditions where administration modality was increased from only injection to include oral administration. Cite the findings (I believe no difference in medication adherence by modality was found).

"potential incentive" (l. 107) does not describe the addition of the administration modality. Replace the phrasing.

Under Methods

Change "interviewed" (l 117, l. 216) to surveyed.

ll. 133-134 – variables are not parametric or non-parametric; tests are such. Correct.

Do not start a sentence with a number.

Under results.

Specify in the  regression Table what is the  dependent variable.

There is no test in the results supporting the conclusion that "needle fear is a determinant of pediatric flu VH". Respondents were asked "Would you prefer to have your son/daughter vaccinated if the route of administration was a nasal spray instead of injection?". This is NOT an intention or a behavior. This is a hypothetical preference. Pls stick to the data.

I did not see a reported test of Table 4. Pls add such a test.

Under conclusions.

There was no change of mind of respondents. Pls stick to the wording of the questions/items.

Needs language editing.

Author Response

(The authors gave the same response as above.)
